# Modelling of Fuel Filter Clogging of B20 Fuel Based on the Precipitate Measurement and Filter Blocking Test

**Imam Paryanto** [1,†]**, Ilyin Abdi Budianta** [1]**, Kanya Citta Hani Alifia** [1]**, Ibnu Maulana Hidayatullah** [1]**, Muhammad Arif Darmawan** [1,2]**, Judistira** [3]**, Tirto Prakoso** [3]**, Antonius Indarto** [3] **and Misri Gozan** [1,*]

1    Department of Chemical Engineering, Faculty of Engineering, Universitas Indonesia, Depok 16424, Indonesia
2    Research Center for Process and Manufacturing Industry Technology, Research Organization for Energy and Manufacture, National Research and Innovation Agency, Jakarta Pusat 10340, Indonesia
3    Department of Chemical Engineering, Faculty of Industrial Engineering, Bandung Institute of Technology, Jalan Ganesha No. 10, Bandung 40132, Indonesia
*    Correspondence: mgozan@ui.ac.id
†    Mr. Imam Paryanto passed away.

**Abstract:** The amount of precipitate and residue affects the timing of fuel filter blockage. This study develops a model for fuel filter blocking based on the Precipitate Measurement. Firstly, a modification of ASTM D 7501 for the Cold Soak Filtration Test (CSFT) measured the amount of precipitate in B20 fuel with variations of soaking temperatures and monoglyceride content in biodiesel. Then, a modified ASTM D 2068 for a filter blocking test (FBT) was conducted to correlate the impurities in the B20 fuel and the clogging limit effects represented by the change of pressure difference and time to reach a pressure drop of 30 kPa. Biodiesel B20 samples were prepared by adding monopalmitin so that each had a monoglyceride value of 0.2%, 0.4%, 0.6%, and 0.8% before blending with petroleum diesel. The modified CSFT showed that the amount of B0 impurity was almost zero. However, the amount of the B20 sample precipitate retained on the filter was higher when a lower soaking temperature and higher monoglyceride content was used in the biodiesel. Similar results in the modified FBT showed that the more impurities, the faster the pressure drop achieved a level of 30 kPa. A much shorter time was needed to reach the pressure drop of 30 kPa for B20 fuel samples with the impurities present in both test powders and precipitate compared to those for the B20 fuel samples with a single type of impurity (either test powders or precipitate). The fuel filter clogging time could also be predicted using the graph of fuel filter clogging time vs. the precipitate weight of B20 fuel derived from the FBT test if the precipitate weight had already been determined by the precipitation test (modified CSFT). The simulation model using Ergun's equation for the FBT of the B20 fuel could also show similar results to that of the FBT experiment, with the difference (averaged errors) ranging from 4.15% to 5.79%.

**Keywords:** biodiesel; B20 fuel; precipitation; fuel filter blocking

## 1. Introduction

Many countries have been promoting using biofuel as a substitute for conventional diesel fuels to reduce the emission of greenhouse gases. Biodiesel is one of several biofuels that has been extensively used in the form of blended fuel. Biodiesel has advantages over petroleum diesel fuel, such as improved emission performance [1], lower sulfur content, a higher flash point, improved lubricity, lower toxicity, and biodegradability [2–4]. Besides its advantages, however, biodiesel still has some technical problems, such as filterability and water–fuel separation [5–7], oxidative stability, low-temperature performance, nitrogen oxides' (NOx) emissions [2,8,9], and its disfavored cold flow characteristics [10,11]. The problem of the low-temperature performance of biodiesel relates to the formation of a precipitate that causes filter clogging [8]; moreover, biodiesel can shorten the durability of engine components, including fuel filters [5,8,12].

Biodiesel is produced by the transesterification reactions of fatty acids from animal or plant oil with alcohol, in which the triglycerides are converted to alkyl esters [3]. However, after esterification, unreacted glycerides, such as mono- or diglycerides, may appear [9]. As impurities, saturated monoglycerides (SMGs) can significantly affect biodiesel even in very low amounts, especially at low temperatures [13]. This is due to the high final melting temperature (FMT) or melting point of SMGs, which will form solid deposits above the cloud point (CP) [14].

There are available tests for assessing the precipitation in biodiesel–petroleum diesel-blended fuel and its clogging effect on fuel filters. ASTM D 7501 for the Cold Soak Filtration Test (CSFT) [15,16] combines the process of cold soaking (in terms of soaking temperatures and time) and filterability to determine fuel filter clogging tendency. Another method is ASTM D 2068 for the filter blocking test (FBT) [17,18], which is a method for determining the fuel filter blocking tendency and filterability of middle distillate fuel oils and non-petroleum liquid fuels such as biodiesel. Previous studies have been carried out for the Cold Soaking Filtration Test of BXX fuels for varied conditions of soaking and the type of monoglyceride [19–23] in palm oil-based biodiesel, and their results showed that the formation of the precipitate was influenced by monoglyceride content in B100, the percentage of B100 in BXX, and soaking temperature conditions.

Fuel filter clogging is closely related to the fuel flow rate and pressure difference before and after the filter because clogging leads to a reduced fluid flow rate and/or increased pressure drop. Studies using filter clogging to model the fluid flow through a porous media date back to the early 1900s. One of the earliest models that related pressure drop to fluid flow was proposed by Forchheimer [24], and his simple model has since been used as a basis for several complex models (e.g., Kozeny–Carman, Ergun, and Endo equations). Darcy's Law is another early model used for calculating the permeability of a filter septum [25,26]. Darcy described the volumetric flow rate of a system as a function of pressure drop, permeability, cross-sectional area to flow, the viscosity of the fluid, and the thickness of the porous medium (e.g., depth of a deep bed filter). The Kozeny–Carman [27] and Ergun [28] equations are two commonly used formulations applied in fluid dynamics to model the pressure drop of a fluid flowing through a porous medium (e.g., packed bed, filter mesh). Further development of clogging filter modelling was then extended by Tien and Ramarao [24], Endo et al. [29], Tien and Bai [30], Ni et al. [31], Liu et al. [32], and Eker et al. [33] to focus on the parameters of porosity, cake thickness/resistance, and pressure drop in the system with various applications and considerations. Even though there have been many studies on filter clogging that focus on the physical modelling of clogging phenomena, there is a lack of usable models that are able to predict filter clogging progression.

The blockage caused by contaminants in the biodiesel synthesis process must be investigated further. This research deals with measuring precipitate in B20 fuel with varied monoglyceride contents in palm oil-based biodiesel and at various soaking temperature conditions, applying a modified CSFT. The new model in this study is expected to provide a more straightforward approach. The results of the precipitate weight retained on the filter are then correlated to the fuel filter blocking tendency applying a modified FBT concerning the time to reach a certain level of the pressure drop across the fuel filter. Mathematical modelling is based on the Precipitate Measurement to describe fuel filter blocking. The model approach we developed is expected to provide a more experimentally accurate perspective of the clogging problem.

## 2. Materials and Methods

### 2.1. Fuel Preparation and Analysis

The biodiesel sample was analysed for several quality parameters according to SNI 7182:2015 [34]. A distilled palm oil-based biodiesel (B100) sample was provided by PT Wilmar Nabati Indonesia in Gresik, East Java, and had a monoglyceride content of 0.174%-mass as in Certificate of Analysis (CoA). Petroleum diesel fuel (B0) used in the

study was taken from PT Pertamina. Monopalmitin was used to increase monoglyceride content in biodiesel for the test purpose. Monopalmitin was obtained from Tokyo Chemical Industry (TCI) Japan, with specifications of melting point 73.0 to 77.0 °C and purity >95% (GC). Increased monoglyceride (MG) content in B100 (for testing) was added by adding monopalmitin to the biodiesel sample so that its MG content reached approximately 0.2, 0.4, 0.6, and 0.8 wt%, respectively, before blending. The monoglyceride contents in biodiesel samples were determined according to ASTM D6584 [35] with Gas Chromatography (Perkin–Elmer Clarus, Waltham, MA, USA), which had a flame ionization detector (GC-FID), an Elite 5-HT column (30 m in length, 0.32 mm internal diameter, and a 0.1 μm film thickness), and a hydrogen gas carrier.

Petrodiesel fuel (B0) was filtered to remove the solid residue before blending to B20 so that only impurities from B100 could be considered as influencing the filter clogging in the B20 precipitation test. B20 samples were prepared by blending B0 and B100 in a volume ratio of 80:20 to produce B20 fuel. To represent the homogenous concentration of impurity in the agitated fuel during the test, the APPIE JIS Test Powders suited to JIS Z 8901 Class 8 [36] were used at a concentration of 0.05 g/L by mixing the powder and fuel. Then, several samples of 5 cc fuel were taken and measured according to their powder concentration.

## 2.2. Precipitation Test

The precipitation test applied modified ASTM D 7501 for the Cold Soak Filtration Test (CSFT). The ASTM D 7501 test method covers the determination by filtration time after cold soak for the suitability of biodiesel fuel blend stock (B100). In the original test method, 300 mL of biodiesel (B100) was stored at 4.5 ± 0.5 °C for 16 h, allowed to warm to 25 ± 1 °C, and vacuum filtered through a single 0.7 μm glass fibre filter at controlled vacuum levels of ~70–85 kPa (21–25 in. of Hg). The filtration time is reported in seconds [15]. This study performed the test by placing 100 mL of each B20 sample in a closed 100 mL-separating funnel. A blank test using B0 was also conducted by filtering B0 through the same-sized filter paper without soaking. The samples were then placed in the refrigerators at a constant, controlled temperature of 20 and 25 °C each and of room temperatures (24–30 °C).

After 21 days of soaking, each sample was vacuum filtered through a filter paper, each having a pore size of 0.8 μm. Cellulose acetate membrane filters used to filter B20 were Sartorious™ with a specification diameter of 47 mm and a particle retention of 0.8 μm. Precipitate retained on filter paper was then washed with petro-ether and dried in a vacuum condition; then, its weight was measured. Petroleum ether from Merck was used to wash the precipitate; therefore, only monoglyceride in the precipitate remained on a filter.

## 2.3. Filter Blocking Test

The fuel filter blocking test applied modified ASTM D 2068 to determine the fuel filter blocking tendency and filterability of middle distillate fuel oils and non-petroleum liquid fuels such as biodiesel. The test applied two types of fuel samples, i.e., (1) 10 L of each clean B0 fuel sample with or without additional test powders impurities, (2) 10 L of each B20 fuel sample with different monoglyceride content in B100 fuel and with or without additional test powders. The 0.5 g of APPIE JIS Test Powders suited to JIS Z 8901 Class 8 [36] was added to 10 L of the fuel sample.

The test was initiated by soaking 10 L of each fuel sample in a 15-litre stainless-steel container at constant temperature conditions of 20 °C, 25 °C, and room temperature for 21 days. Each fuel sample was pumped at a constant flow rate of 0.2 litre/minute to pass a fuel filter of 90 mm in diameter and with a 0.8-micron pore size. The pressure drop in the filter was measured at an increment time of 1 s. If the pressure drop reached a maximum pressure drop of 30–45 kPa or the experiment lasted for 40 min (due to the low level of the final fuel quantity to be pumped), the test was stopped. The modified FBT system is described in Figure 1.

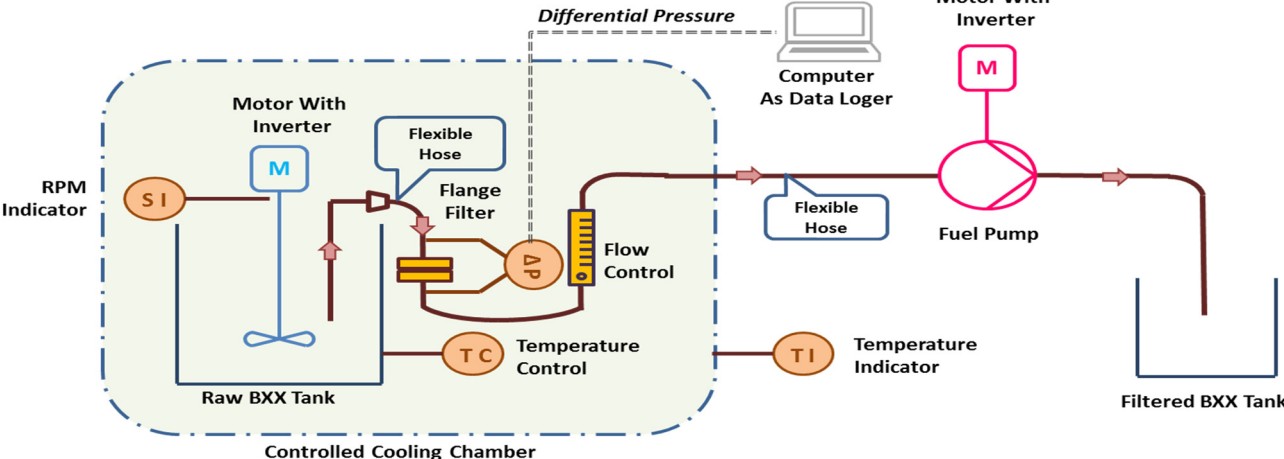

**Figure 1.** Modified FBT system.

The homogeneity of the powder impurity in the fuel sample was checked by taking samples of 5 mL of B20 fuel stirred in the Raw B20 Tank. Those samples were then measured for the number of impurities using the method described in the Precipitation Test Procedure.

### 2.4. Mathematical Modelling of Fuel Filter Blocking

The mathematical modelling was constructed by observing the pressure drop change in the fuel filter as fuel passed through the filter. The change in pressure drop was caused by the accumulation of fuel impurities blocking the fuel flow from passing through the filter's pore. The fuel flow passing through the fuel filter depends on the fuel impurities, as described in Figure 2a,d.

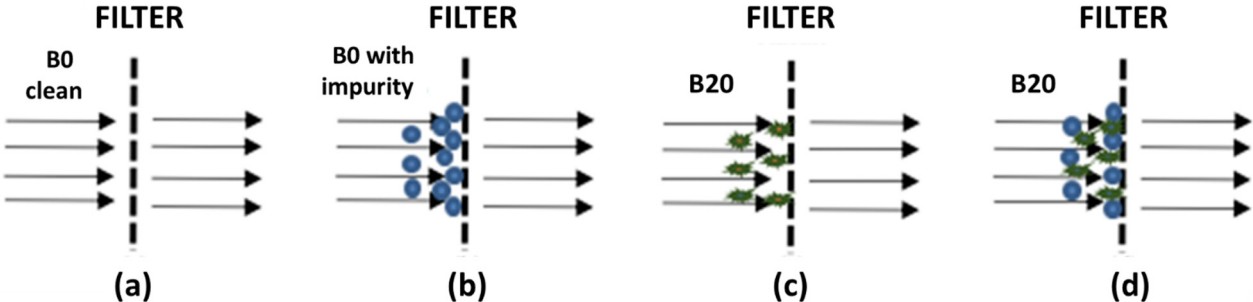

**Figure 2.** The flow of fuels containing different types of impurities: (**a**) clean B0 fuel, (**b**) solid test powder impurities in the B0 fuel, (**c**) precipitate impurity in B20 fuel, and (**d**) mixed test powder and precipitate impurities in B20 fuel.

Figure 2a shows that clean liquid fuel passed through the fuel filter without blocking its flow due to no impurities. Figure 2b illustrates different phenomena in which the solid test powder impurities in B0 or B20 fuel blocked the fuel's flow through the filter pores. However, since there were void spaces among the solid "dust", the liquid fuel could still pass through the filter pore. Nevertheless, there would be increased pressure drops as the accumulated solid "dust" became a packed solid bed with some void spaces among the solid "dust". Figure 2c showed that waxy, non-rigid precipitate in B20 fuel could block the fuel flow as it accumulated and remained on the filter. Because of the non-rigid (flexible) form of waxy precipitate, as the pressure drop increased, the liquid fuel could still pass through the fuel filter, but again with increased pressure drops. A different case was found in Figure 2d, where two types of impurities were present in the fuel and accumulated on the filter. The void spaces previously present, as in Figure 2b, were filled with waxy precipitate,

forming a very packed bed. This condition could lead to a dramatically increased pressure drop in the fuel filter.

The mathematical modelling used Darcy's Law to describe the phenomena in Figure 2b. The Darcy equation is a model used to calculate the permeability of a filter septum. Filtration with the formation of a packed bed (cake) on the filter could be written as a differential equation using a modified Darcy equation [25,26,33], as shown in Equation (1).

$$\frac{dV}{A\,d\theta} = \frac{K\Delta P}{\mu L} \tag{1}$$

The volumetric flow rate '$V$' of a system as a function of pressure drop '$\Delta P$', permeability '$K$', cross-sectional area to flow '$A$', viscosity '$\mu$' of the fluid, and the thickness '$L$' of the porous medium (e.g., depth of a packed bed filter).

The pressure drop change at an elapsed time for Figure 2c,d could follow the Ergun equation with the proposed assumption that the accumulated impurities blocked the filter pore as a packed bed, which depends on the superficial velocity ($V_s$), viscosity ($\mu$), porosity ($\epsilon$), packed bed thickness ($L$), and average diameter ($D_P$) of the impurities in the fuel. The change of pressure drop for a certain time could be correlated to the Ergun equation, which has been derived by Liu et al. [32,33], as follows:

$$\Delta P = \frac{10\,A\,V_S\,\mu\,(1-\epsilon)^2\,L}{D_P{}^2\,\epsilon^3} + \frac{B\,(1-\epsilon)\,\rho\,V_S{}^2\,L}{D_P\,\epsilon^3} \tag{2}$$

Specifications for each parameter stated in Equation (2) are described in Table 1.

**Table 1.** Specification of parameters stated in Ergun equation.

| Parameter | Value | Units |
|---|---|---|
| Density of B20 fuel ($\rho$) | 848.30 | $kg/m^3$ |
| Kinematic viscosity of B20 fuel ($\mu$) | 2.49 | $mm^2/s$ |
| Flow rate of B20 fuel ($V_s$) | 0.20 | L/min |
| Concentration of added impurity "powder" JIS 8 | 0.05 | g/L |
| Impurity powder diameter (APPIE, Japan) [36] | 7.60 | micron |
| Impurity powder density (APPIE, Japan) [36] | 3.00 | $g/cm^3$ |
| Filter paper diameter (paper) | 90.00 | mm |
| Filter porosity (paper) | 0.80 | micron |
| Precipitate diameter | 5.30 | micron |

## 3. Results and Discussion

### 3.1. Preparation of B20 Fuel

Biodiesel was analysed for several quality parameters according to SNI 7182:2015 [34], as shown in Table 2. All were within biodiesel quality standards according to SNI 7182:2015 and ASTM D6571 specifications. The original monoglyceride content in biodiesel before the addition of monopalmitin was determined to be 0.179%-mass, containing 0.094%-mass monopalmitin, 0.070%-mass monoolein, and 0.015%-mass monostearin. Monopalmitin and monostearin are saturated monoglycerides, whereas monoolein is an unsaturated monoglyceride. Monopalmitin was added to biodiesel to vary monoglyceride content in biodiesel samples to influence precipitate formation in BXX fuel. The modified monoglyceride contents of the biodiesel samples were 0.437%, 0.623%, and 0.824%. By adding monopalmitin, the percentage of the saturated monoglyceride (SMG) composition, i.e., monopalmitin and monostearin, changed from an initial percentage of 60.9%-mass to 83.8, 88.6, and 91.4%-mass, respectively.

**Table 2.** Biodiesel fuel (B100) specification and quality.

| No | Parameter | Unit | B100 Sample | | Limit SNI 7182-2015 | | Methods |
|----|-----------|------|-------------|---|---------------------|---|---------|
| | | | CoA | Result | Min | Max | |
| 1 | Density at 40 °C | kg/m3 | n/a | 855.3 | 850 | 890 | SNI 7182-2015/ASTM D 4052 |
| 2 | Cloud point | °C | 15 | 16 | | 18 | ASTM D 2500 |
| 2 | Ester content | % mass | n/a | 98.7 | 96.5 | | Calculated |
| 3 | Free glycerol | % mass | 0.003 | 0.006 | | 0.02 | ASTM D 6584 |
| 4 | Total glycerol | % mass | 0.047 | 0.128 | | 0.34 | ASTM D 6584 |
| 5 | Monoglyceride: | % mass | 0.174 | 0.179 | | 0.8 | EN 14105/ASTM D 6584 |
| | a. Monopalmitin | % mass | | 0.094 | | | |
| | b. Monoolein | % mass | | 0.070 | | | |
| | c. Monostearin | % mass | | 0.015 | | | |

The samples B20 were prepared by blending B0 and B100 as explained in Section 2.1. B20 samples were analysed for several related quality parameters, as shown in Table 3 and Figure 3.

**Table 3.** Monoglyceride content variation in biodiesel samples.

| No | Biodiesel Samples | Added Monopalmitin, mg/L B100 | Monoglyceride Content %-mass | Monopalmitin %-mass | Monoolein %-mass | Monostearin %-mass |
|----|-------------------|-------------------------------|------------------------------|---------------------|------------------|--------------------|
| 1 | B100-MG initial | - | 0.179 | 0.094 | 0.070 | 0.015 |
| 2 | B100-MG 0.4 | 2038.76 | 0.437 | 0.348 | 0.071 | 0.019 |
| 3 | B100-MG 0.6 | 3842.97 | 0.623 | 0.533 | 0.071 | 0.019 |
| 4 | B100-MG 0.8 | 5647.18 | 0.824 | 0.732 | 0.071 | 0.021 |

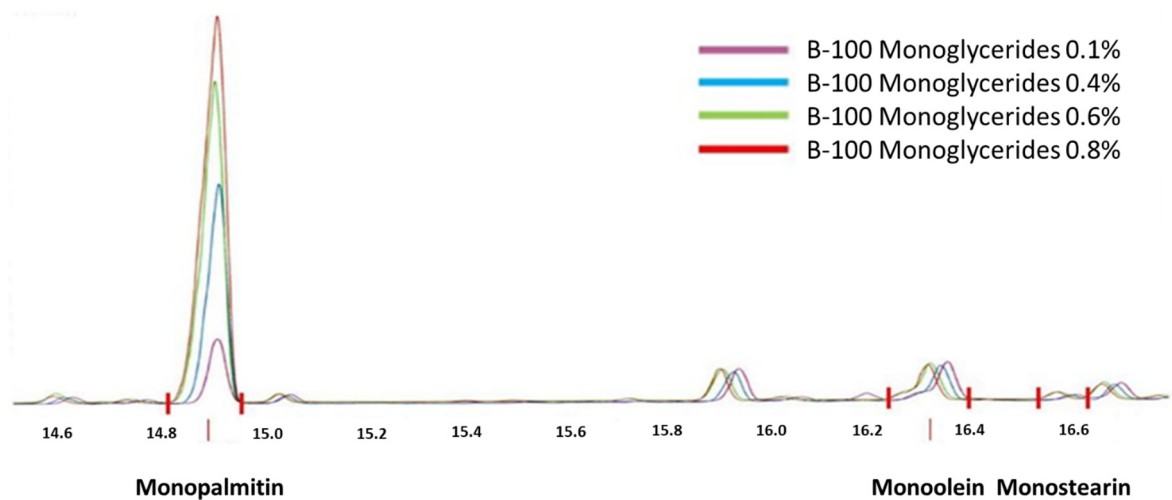

**Figure 3.** GC Chromatogram for monoglyceride content in biodiesel.

The result for some important quality parameters of petrodiesel fuel (B0) was shown in Table 4, whereas the results of some important quality parameters of B20 fuel were presented in Table 5. Both results showed that the selected quality parameters of B0 and B20 fuels had met the standard of ASTM.

**Table 4.** Petrodiesel fuel (B0) specification and quality.

| No | Parameter | Unit | Result | Standard B0 (Solar 48) | | Methods |
| --- | --- | --- | --- | --- | --- | --- |
| | | | | Min | Max | |
| 1 | Density at 15 °C | $kg/m^3$ | 843.8 | 815 | 860 | ASTM D4052 |
| 2 | Kinematic viscosity at 40 °C | $mm^2/s$ | 2.6 | 2 | 4.5 | ASTM D445 |
| 3 | Cloud point | °C | 9.7 | - | 18 | ASTM D5773 |
| 4 | Sulfur content | % mass | 0.106 | | 0.25 | ASTM D4294 |

**Table 5.** B20 specification and quality.

| No | Parameter | Unit | Result (B20) | Limit | Methods |
| --- | --- | --- | --- | --- | --- |
| 1 | Density at 15 °C | $kg/m^3$ | 848.30 | 815–860 | ASTM D4052 |
| 2 | Kinematic viscosity at 40 °C | $mm^2/s$ | 2.98 | 2.0–4.5 | ASTM D445 |
| 3 | Cloud point | °C | 9.70 | 18 max | ASTM D5773 |
| 4 | Water content | %-vol | 249.87 | 500 max | ASTM D6304 |
| 5 | Sediment content | %mass | None | 0.01 max | ASTM D473 |
| 6 | FAME content | %mass | 20.10 | - | ASTM D7806 |
| 7 | Total acid number | mg KOH/g | 0.089 | 0.06 | ASTM D664 |
| 8 | Oxidation stability (Rancimat) | Hours | 36.21 | 35 | EN 15751 |

### 3.2. Results of Precipitation and Filter Blocking Tests

After soaking each sample of B20 fuel in three different soaking temperature conditions, the precipitate of B20 fuel was formed. The B20 filtration tests (Table 6) showed that biodiesel's lower soaking temperature conditions and higher monoglyceride content caused further precipitation. This result is confirmed by the previous result [19,20]. It was shown in Table 6 that the precipitate amount of B20 fuel with B100 having a monoglyceride content of 0.2% or below for all soaking temperature conditions had the lowest amount, and this low amount of precipitate might not have an effect on the filter blocking.

**Table 6.** Result of precipitation test.

| Samples | Weight of Precipitate (g/100 mL) at Soaking Temperatures | | | | | | | | |
| --- | --- | --- | --- | --- | --- | --- | --- | --- | --- |
| | 20 °C | | | 25 °C | | | Room Temp (26–30 °C) | | |
| B20 | Batch 1 | Batch 2 | Average | Batch 1 | Batch 2 | Average | Batch 1 | Batch 2 | Average |
| B20-with B100 0.179%MG | 0.0012 | 0.0013 | 0.0013 | 0.0014 | 0.0012 | 0.0013 | 0.0012 | 0.0011 | 0.0012 |
| B20-with B100 0.20%MG | 0.0033 | 0.0036 | 0.0035 | 0.0021 | 0.0023 | 0.0022 | 0.0027 | 0.0014 | 0.0021 |
| B20-with B100 0.4%MG | 0.0125 | 0.0114 | 0.0120 | 0.0067 | 0.0073 | 0.0070 | 0.0051 | 0.0061 | 0.0056 |
| B20-with B100 0.6%MG | 0.0278 | 0.0246 | 0.0262 | 0.0142 | 0.0153 | 0.0148 | 0.0099 | 0.0105 | 0.0102 |
| B20-with B100 0.8%MG | 0.0400 | 0.0414 | 0.0407 | 0.0256 | 0.0231 | 0.0244 | 0.0192 | 0.0183 | 0.0188 |
| B0 | 0.0012 | 0.0011 | 0.0012 | 0.0012 | 0.0014 | 0.0013 | 0.0009 | 0.0011 | 0.0010 |
| B20 Market | 0.0228 | 0.0239 | 0.0234 | 0.0201 | 0.0219 | 0.0210 | 0.0196 | 0.0202 | 0.0199 |

The fuel filter blocking test result was described in Figures 4 and 5. From those two graphs, it could be explained that the more precipitate was formed, the faster the pressure drop increased. FBT using B0 fuel without added impurity showed no increase in pressure drop, while those using the B0 fuel with added impurity powder indicated a constant increase in their pressure drops. The built-up impurity powder retained and accumulated on the filter caused the pressure drop to increase as the fuel passed through the filter (also referred to in Figure 2b). The fuel could still flow through the void of the packed bed of the impurity powder and then passed through the filter.

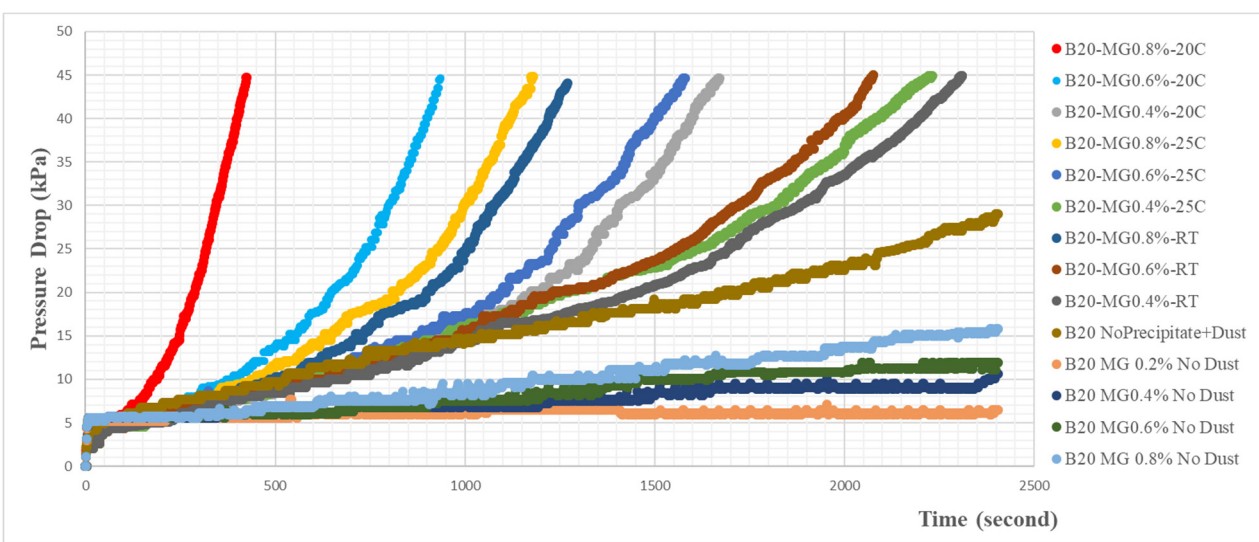

**Figure 4.** Graph of pressure drop trend for the different conditions of soaking temperatures and monoglyceride content in B20 fuel.

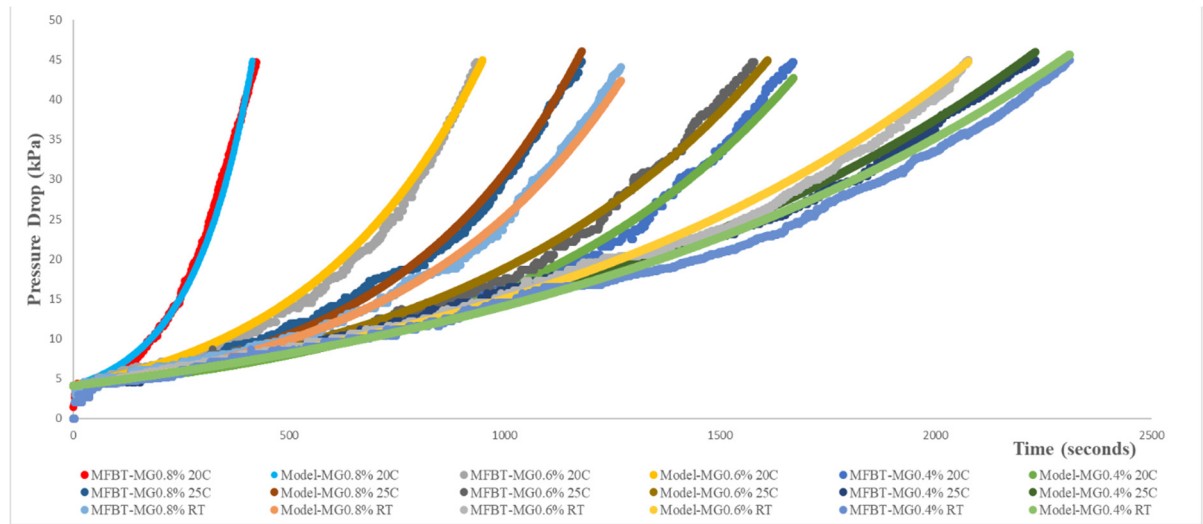

**Figure 5.** Pressure drop trend resulting from modified FBT test and modelling.

When the B20 fuel with monoglyceride impurities was pumped through the fuel filter, there was a slight increase in the pressure drop. The flexible form of waxy monoglyceride precipitate might accumulate on the filter that could still allow the fuel to pass through the fuel filter (Figure 2c). The blocking effect of the flexible form of waxy precipitate was less than the packed bed of impurity powder due to the rigidity of the packed, accumulated impurity powders, allowing the fuel only to flow through the voids of the packed impurity powders.

The flexible form of the waxy impurity allowed the movement of the waxy impurity so that the fuel could still pass through the accumulated flexible waxy impurity more freely compared to the void of packed accumulated solid powders.

The severity of the blocking effect of the impurity increased if the combined solid and flexible waxy impurities were present in the B20 fuel. The waxy impurities could gradually fill the voids (Figure 2d), as previously described for the packed bed solid impurity (powders). This condition could lead to a dramatically increased pressure drop in the fuel filter. This explains why the combined solid and waxy impurities could not produce the sum of each effect of the increased pressure drop. Figure 4 shows that there

were significant pressure drop increases if both impurities were present in the B20 fuel. The blocking effects were even worse if the quantity of the waxy precipitate was increased.

### 3.3. Results of Modelling of Fuel Filter Blocking

Mathematical modelling was constructed to describe the filter-blocking effect for the phenomena explained above. Based on Equation (2), the form of modelling was categorized into two models; namely, the first model (Model 1) was used to explain the filter blocking referred to in Figure 2c, and the second model (Model 2) referred to Figure 2d. Due to the effect of the different quantities of waxy precipitate in the B20 fuel on the filter blocking because of the different soaking temperature conditions during the sample preparations, the second model calculation was divided into two models, namely, Model 2A for soaking temperature conditions of 20 °C (producing a higher amount of precipitate content in the fuel) and Model 2B for temperature conditions of 25 °C and room temperature.

After one minute of the running test, the initial point was set to simplify the modelling equation in Figures 4 and 5. This setting allows the accumulated impurities retained on the filter to produce its blocking effect through a pressure difference change from about 5 kPa up to 45 kPa or for 40 min of the running test. Therefore, the modelling started when there was a stable change in pressure drop before it increased significantly.

It was assumed that the void porosity of the accumulated precipitate was 0.371 because all precipitate particles were the same size, uniform in shape, and regularly packed. The change in the packed bed thickness could be modelled into an equation as a function of the time and concentration of accumulated precipitate impurities remaining on the filter. The rate of the pressure drop increase as a function of the increase of the packed bed thickness could be linearly modelled into Equation (3).

$$\frac{dL}{dt} = K_1 \left( \frac{n \, V_P}{V_{filter}} \right) \left( \frac{D_P}{D_{Pori}} \right) D_P$$

$$\frac{dP}{dt} = K_1 \left\{ \frac{10 \, A \, V_S \, \mu \, (1-\epsilon)^2}{D_P^2 \, \epsilon^3} + \frac{B \, (1-\epsilon) \, \rho \, V_S^2}{D_P \, \epsilon^3} \right\} \left( \frac{n \, V_P}{V_{filter}} \right) \left( \frac{D_P}{D_{void}} \right) D_P \qquad (3)$$

$$n = 10 \, (1-\epsilon) \left( \frac{C_{particle} \, V_{filter}}{m_{particle}} \right)$$

In the above equation, $K_1$ is a constant, $D$ diameter, $V$ volume, and $n$ is the number of accumulated precipitate impurities. $V_{filter}$ was the filter volume calculated using the length value as much as the particle diameter. The model simulation was carried out by using the value of each component on the equation, using Table 7. The equation of the packed bed thickness change was then substituted into the Ergun equation, which was used and simulated for 40 min of the test or until the pressure drop reached 45 kPa.

**Table 7.** Equation model for the change of packed bed thickness (Models 2A and 2B).

| Model 2A | | Model 2B | |
|---|---|---|---|
| $\frac{dL}{dt} = K_2(n_1 + n_2)$ | Equation (4) | $\frac{dL}{dt} = K_3(1-\epsilon)$ | Equation (5) |
| $n = (1-\epsilon)\left(\frac{C_{particle} \, V_{filter}}{m_{particle}}\right)$ | Equation (6) | | |

To describe the filter blocking effect referred to in Figure 2d, the model for the packed bed change in terms of its thickness and void porosity could be categorized into two models (Models 2A and 2B), as described earlier. In these two models, the waxy precipitate and added powder impurities acted as blocking particles. Because of the difference in the physical and flexibility form between them, the equation describing the change of packed bed thickness could be a function of the time of running the test and the concentration of accumulated blocking particles, which caused its combined effect due to their different

physical forms. The correlation of different forms of both impurity types blocking the filter could significantly affect the change of pressure drops in the filtering process. Supposing that the combined solid and flexible waxy impurities were present in the B20 fuel, this could be caused by the voids previously described for the packed bed solid impurity (powders) in Figure 2b, which were gradually filled with the waxy impurity (Figure 2d), since the amount of the powder and impurity concentration were added at a constant value. Thus, the amount of waxy impurity (which depends on the monoglyceride content in the B20 fuel and the soaking temperatures) could influence the change in pressure drops. This condition could dramatically change the pressure drop at the fuel filter. Therefore, the differential equation of the change in packed bed thickness could be modified for this condition by adding new constants, namely $K_2$ for Model 2A (Equations (4) and (6)) and $K_3$ for Model 2B (Equation (5)), as described in Table 7.

The changes in the $K_2$ and $K_3$ constants were caused by the presence and interaction of both forms of solid and flexible waxy impurities, which affected the thickness and porosity values of the packed bed on the filter. The porosity value could decrease as the thickness of the packed bed of accumulated mixed impurities increased. One of the constants in the equation became the function of the value of the impurity concentration ($C_{partiel1}$). The rate in the change of packed bed porosity could follow the equation below (Equation (7)).

$$\frac{d\epsilon}{dt} = -\left( F(x) \left( K_4 \left( t^{(K_5(t^{K_6}))} + t^{K_7} \right) \right) \right) \tag{7}$$

The equation $F(x)$ for Model 2A was formulated in Equation (8).

$$F(x) = 10^{-5} \left( \frac{C_{particle\ 1}}{C_{particle\ 2}} \right)^3 - 6 \times 10^{-4} \left( \frac{C_{particle\ 1}}{C_{particle\ 2}} \right)^2 + 1.04 \times 10^{-2} \left( \frac{C_{particle\ 1}}{C_{particle\ 2}} \right) + 0.9869 \tag{8}$$

The equation $F(x)$ for Model 2B was set as in Equation (9).

$$F(x) = -0.0198 \left( \frac{C_{particle\ 1}}{C_{particle\ 2}} \right)^4 + 0.1805 \left( \frac{C_{particle\ 1}}{C_{particle\ 2}} \right)^3 - 0.4602 \left( \frac{C_{particle\ 1}}{C_{particle\ 2}} \right)^2 + 0.5233 \left( \frac{C_{particle\ 1}}{C_{particle\ 2}} \right) + 0.2928 \tag{9}$$

The $K_2$, $K_3$, $K_4$, $K_5$, $K_6$, and $K_7$ were constants in those equations. $C_{particle\ 1}$ belongs to waxy precipitate particles, and $C_{particle\ 2}$ to solid powder particles. Then, the equation for the packed bed thickness changes, and the bed's void porosity for fuels without added powder impurity is substituted into the Ergun equation and simulated for 440 s for the B20 fuel. However, for B20 fuel samples with different monoglyceride content and various temperature soaking conditions, the time taken to run the tests can be shown in Table 8. The rate of the pressure drop increase as a function of the increased thickness of the packed bed and the decrease in the void porosity of the bed could follow the equation by Swanson et al., 2016, by substituting the equation of the packed bed thickness change ($dL/dt$) and the change of the void porosity ($d\epsilon/dt$). The solution of those equations was carried out by applying a software called FlexPDE ver. (PDE Solutions Inc., Washington, DC, USA) 7. The simple equation form for the changes of pressure drop is shown in Equation (10).

$$\frac{dP}{dt} = \frac{10\,A\,V_S\,\mu}{D_P{}^2\,\epsilon^3} \left( \frac{\epsilon(1-\epsilon)^2 \frac{dL}{dt} - (1-\epsilon)(3-\epsilon)L\frac{d\epsilon}{dt}}{\epsilon} \right) + \frac{B\,\rho\,V_S{}^2\,L}{D_P\,\epsilon^3} \left( \frac{(2\epsilon-3)L\frac{d\epsilon}{dt}}{\epsilon} + (1-\epsilon)\frac{dL}{dt} \right) \tag{10}$$

**Table 8.** Simulation time for each sample of B20 fuel with added powder impurity.

| B20 Sample Description | Simulation Time (Seconds) |
|---|---|
| B20–MG 0.80%–20 °C | 416 |
| B20–MG 0.60%–20 °C | 950 |
| B20–MG 0.40%–20 °C | 1671 |
| B20–MG 0.80%–25 °C | 1180 |
| B20–MG 0.60%–25 °C | 1611 |
| B20–MG 0.40%–25 °C | 2231 |
| B20–MG 0.80%–room temperature | 1270 |
| B20–MG 0.60%–room temperature | 2077 |
| B20–MG 0.40%–room temperature | 2311 |

After simulating the model equation, the values of constants were found, as shown in Table 9.

**Table 9.** The values of constants of the simulation model.

| Constants | | Value |
|---|---|---|
| $A$ | | $2671 \times 10^{6}$ |
| $B$ | | $1534 \times 10^{2}$ |
| $K_1$ | | $6112 \times 10^{-4}$ |
| $K_2$ | | $7527 \times 10^{-17}$ |
| $K_3$ | | $9625 \times 10^{-12}$ |
| $K_4$ | | $3171 \times 10^{-5}$ |
| $K_5$ | Model 2A | $1280 \times 10^{-4}$ |
| | Model 2B | $5110 \times 10^{-5}$ |
| $K_6$ | Model 2A | 0.713 |

Using these constants for Equations (2) to (7), the graphs for these models for representing fuel filter blocking are shown in Figure 5.

The differences (averaged errors) between the pressure drop values from the modified FBT test and those from the simulation model are described in Table 10 and ranged from 4.15% to 5.79%.

**Table 10.** The difference (averaged errors) between the pressure drop values from the modified FBT test and those from the simulation model.

| B20 Sample | Difference (Average Errors) % |
|---|---|
| B20-MG0.8%-20C | 4.91 |
| B20-MG0.6%-20C | 5.79 |
| B20-MG0.4%-20C | 4.68 |
| B20-MG0.8%-25C | 5.42 |
| B20-MG0.6%-25C | 5.30 |
| B20-MG0.4%-25C | 5.27 |
| B20-MG0.8%-RT | 4.15 |
| B20-MG0.6%-RT | 4.56 |
| B20-MG0.4%-RT | 4.67 |
| Average | 4.97 |

It was also found that the time to reach the pressure drop of 30 kPa for each B20 fuel sample, for both the modified FBT test and the simulation model, was influenced by the amount of precipitate present in the B20 fuel. This explanation corresponds with Figure 6, as shown below. Therefore, the more precipitate that was present in the B20 fuel, the shorter the pressure drop reached 30 kPa. In other words, the large amount of precipitate found in the B20 fuel could cause faster fuel filter clogging. The graph in Figure 6 can predict the

time to reach a pressure drop of 30 kPa if the amount of precipitate was known and had already been determined by the precipitation test method.

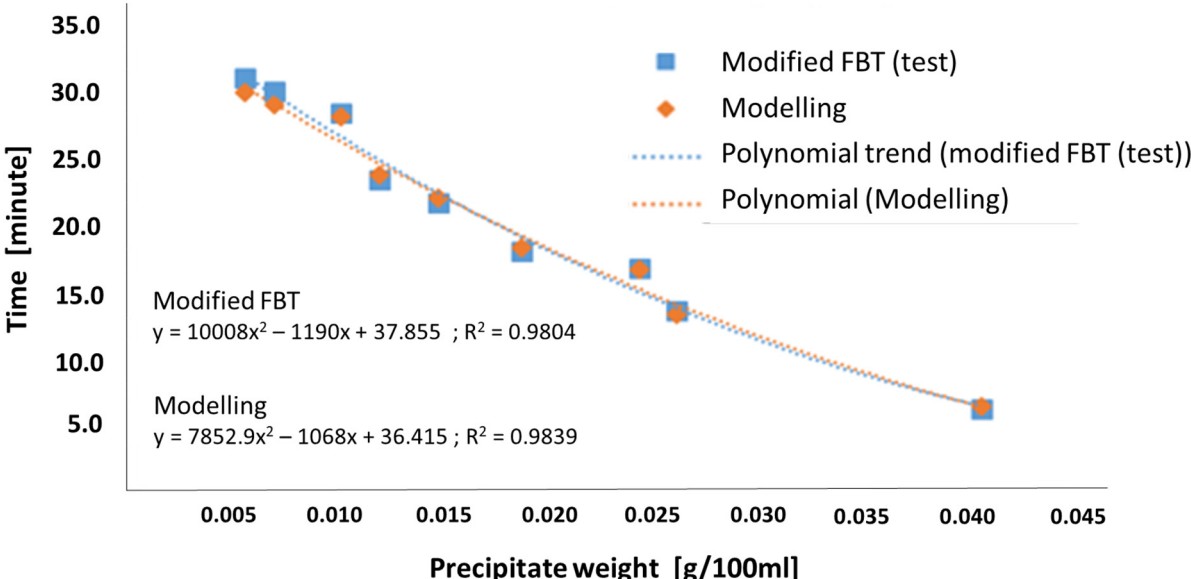

**Figure 6.** The time to reach pressure drop of 30 kPa vs. precipitate weight from modified FBT test and modelling.

The method which used monopalmitin in the simulated monoglyceride was the one with the highest composition and the highest tendency for the clogging filter to have a melting point of 65–68 °C (Komariah et al., 2018). The current highest application of biodiesel blends worldwide is B20, such as that used in Indonesia, Costa Rica, and Minnesota. Other countries even still apply B2 to B10. Thus, B20, with the highest monopalmitin base, represents the highest potential risk currently available. This paper uses palm-based biodiesel, wherein the cloud point is at 16 °C. Therefore, the experiment carried out at the lowest temperature of 20 °C is the most appropriate, and the model is also not built for biodiesel with cloud points below 16 °C. From these considerations, this model is the most suitable for palm oil biodiesel because it can be used to provide knowledge of the blockage's characteristics in tropical climate conditions.

## 4. Conclusions

From the precipitation test conducted using B20 fuel samples in different soaking temperature conditions, it can be concluded that the amount of precipitate formed in B20 fuel was affected by soaking temperature conditions and monoglyceride content in biodiesel. The higher the precipitate, the lower the soaking temperature and the higher the monoglyceride content in biodiesel. The modified filter blocking test of ASTM D 2068 showed that the B20 fuel, produced from B0 fuel with added solid impurities, having more precipitate, tended to have the fastest filter-blocking effect, i.e., faster to reach the pressure drop of 30 kPa. The blocking effect of the combined solid and flexible waxy impurities was more severe than that of only the flexible waxy impurities in the B20 fuel. The combined solid and flexible waxy impurities in B20 fuel could cause the voids, previously described for the packed bed solid impurity (powders), to be gradually filled with the waxy impurity. This condition could lead to a dramatically increased pressure drop in the fuel filter. The fuel filter clogging time could be predicted using the graph of fuel filter clogging time vs. the precipitate weight of B20 fuel derived from the FBT test. The prediction of time to reach a pressure drop of 30 kPa applies if the precipitate weight has already been determined by the precipitation test (modified CSFT). The simulation model using the Ergun equation for the FBT of B20 fuel could also show similar results to the FBT experiment, with the difference (averaged errors) ranging from 4.15% to 5.79%.

**Author Contributions:** Conceptualization, I.P., T.P. and M.G.; Methodology, I.P. and M.G.; Software, I.P., A.I. and J.; Validation, J., I.M.H. and T.P.; Formal Analysis, I.A.B., K.C.H.A. and I.M.H.; Investigation, I.A.B. and K.C.H.A.; Writing original draft, I.P., M.A.D. and M.G.; Writing review and editing, M.A.D. and M.G.; Supervision, T.P. and M.G., Visualization, A.I. and J.; Project administration, I.A.B. All authors have read and agreed to the published version of the manuscript.

**Funding:** This research was supported by The Ministry of Research and Higher Education of Indonesia through the research grant "*Hibah Penelitian Dasar Unggulan Perguruan Tinggi*" (PDUPT) 2021 (NKB-854/UN2.RST/HKP.05.00/2022). The authors also gratefully acknowledged the financial support for publication by Universitas Indonesia through the scheme of "*Hibah Publikasi Terindeks Internasional*" (PUTI) Q1 2022 (NKB-1151/UN2.RST/HKP.05.00/2022).

**Institutional Review Board Statement:** Not applicable.

**Informed Consent Statement:** Not applicable.

**Conflicts of Interest:** The authors declare no conflict of interest.

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
