# Peer review of "Modelling of Fuel Filter Clogging of B20 Fuel Based on the Precipitate Measurement and Filter Blocking Test"

_2305-7084, doi:10.3390/chemengineering6060084_

Round 1

Reviewer 1 Report

The proposed model considers the results only for the B20 type ester. In addition, the lowest test temperature is 20C, representing a small part of the globe. The described parameters limit the simulation model very much, which should be tested in a wide range. This would allow it to be considered universal and widely useful.

List of mistakes:

Line 110 - The authors mention table 3, but this table is on line 217, 3 pages later. Why is the first table in the text number 3?

Line 215 – Table 2 is on two pages.

Line 224 – Table 4 is on two pages.

Line 362 – Table 9 is on two pages.

Line 372 – Table 10 is on two pages.

Line 384 – Quality of figure 6.

Questions/Suggestions:

It would be a good idea to broaden the scope of the research to verify the proposed model.

Conclusions

The work is mainly written clearly, and the proposed model has potential, but the model is developed for specific boundary conditions, so nothing can be said about its universality at the moment.

Reviewer 2 Report

I have reviewed the manuscript entitled "Modelling of Fuel Filter Clogging of B20 Fuel Based on the Precipitate Measurement and Filter Blocking Test" by Imam Paryanto et al. for publication in ChemEngineering. The authors develop a model for fuel filter blocking based on the Precipitate Measurement. There are some points in the work as written that need to be revised, so I recommend it for publication after major revisions.

-The novelty of the work is not clear. I suggest highlighting more the novelty and the relevance of the work done than what is already in the literature and improve the introduction and the discussion of results.

-Improve the introductory part by more fully describing the biodiesel production process and avoiding unnecessary repetition of the same concept.

-Improve the quality of figures 2,3, and 6.

Reviewer 3 Report

The manuscript "Modelling of Fuel Filter Clogging of B20 Fuel Based on the Precipitate Measurement and Filter Blocking Test" is found good.

The quantum of work did present by the author is sufficient

Validation of process and results should be added.

Reviewer 4 Report

Author shoud add or do the necessary actions as below

1. you should give details of monopalmitin addition, procedure. Is ither any other additive, why you have slectd specifically monoplmitin?

2. you should give details of cold soat filtration test

Round 2

Reviewer 1 Report

Thank you for the answers and corrections to the text of the article. Despite some shortcomings, I am satisfied with the present form of the article.

Reviewer 2 Report

The authors appropriately edited the manuscript according to the advice of the reviewers.